# Exploring Extracellular Vesicle Surface Protein Markers Produced by Glioblastoma Tumors: A Characterization Study Using In Vitro 3D Patient-Derived Cultures

**DOI:** 10.3390/cancers16223748

**Published:** 2024-11-06

**Authors:** Sara Franceschi, Francesca Lessi, Mariangela Morelli, Michele Menicagli, Paolo Aretini, Carlo Gambacciani, Francesco Pieri, Gianluca Grimod, Maria Grazia Trapanese, Silvia Valenti, Fabiola Paiar, Anna Luisa Di Stefano, Orazio Santo Santonocito, Francesco Pasqualetti, Chiara Maria Mazzanti

**Affiliations:** 1Fondazione Pisana per la Scienza, 56017 Pisa, Italy; f.lessi@fpscience.it (F.L.); m.morelli@fpscience.it (M.M.); m.menicagli@fpscience.it (M.M.); p.aretini@fpscience.it (P.A.); c.mazzanti@fpscience.it (C.M.M.); 2Department of Neurosurgery, Spedali Riuniti di Livorno, 57124 Livorno, Italy; carlo.gambacciani@uslnordovest.toscana.it (C.G.); francesco.pieri@uslnordovest.toscana.it (F.P.); gianluca.grimod@uslnordovest.toscana.it (G.G.); annaluisa.distefano@uslnordovest.toscana.it (A.L.D.S.); orazio.santonocito@uslnordovest.toscana.it (O.S.S.); 3Department of Radiation Oncology, Azienda Ospedaliera Universitaria Pisana, University of Pisa, 56126 Pisa, Italy; 29558429@studenti.unipi.it (M.G.T.); 25652525@studenti.unipi.it (S.V.); fabiola.paiar@unipi.it (F.P.); 4Department of Radiation Oncology, Istituto Oncologico Veneto IOV—IRCCS, 35128 Padova, Italy; francesco.pasqualetti@oncology.ox.ac.uk; 5Department of Surgery, Oncology and Gastroenterology, University of Padova, 35121 Padova, Italy

**Keywords:** glioblastoma, tumor-derived explants, extracellular vesicles, surface markers, tumor microenvironment

## Abstract

Glioblastoma is a highly aggressive brain cancer with limited treatment options. This research aims to improve our understanding of GBM progression by focusing on extracellular vesicles released by tumor cells that carry important biological information. The authors developed a laboratory model that closely mimics the tumor microenvironment, allowing for the study of extracellular vesicles released from various cell types within the tumor. This model enabled the identification of specific surface markers on extracellular vesicles that could serve as potential noninvasive diagnostic tools for GBM. The findings suggest that these markers may help improve extracellular vesicle isolation and provide insights into the molecular characteristics of the tumor. This study has the potential to significantly impact the research community by offering new approaches to GBM diagnosis and advancing our knowledge of tumor biology, which could ultimately inform future therapeutic strategies.

## 1. Introduction

Glioblastoma (GBM) is an aggressive form of brain cancer with an extremely poor prognosis, underscoring the urgent need for new diagnostic and therapeutic approaches [1]. Recent studies have explored the potential of GBM-derived extracellular vesicles (EVs), which are lipid bilayer, membrane-bound vesicles secreted by cells and involved in intercellular communication [2]. EVs are complex carriers of microRNAs, proteins, and even DNA mutations, and their cargo holds promise as a source of biomarkers for GBM diagnosis, treatment monitoring, and possibly even prognosis prediction [3,4,5,6].

EVs encompass a range of subtypes, including exosomes, microvesicles, and apoptotic bodies, each of which varies in size, biogenesis pathway, and cargo composition [7,8]. The identification and functional characterization of EV subtypes remain challenging due to their overlapping characteristics, difficulties in isolating specific subtypes, and the lack of methods for conclusively demonstrating subcellular origin [9,10]. As outlined in the latest guidelines from the International Society for Extracellular Vesicles (MISEV2023) [11], EVs are defined as lipid bilayer-bound particles released from cells that cannot replicate, given the absence of a functional nucleus. These updated guidelines also offer recommendations to address key challenges associated with EV research, such as standardized methods for EV isolation, characterization, and functional studies [12].

Moreover, the unique surface markers of these EVs not only help to identify them, but also offer information about their origin, content, and function [13,14]. This knowledge is critical for unlocking targeted therapies and promoting advances in cancer biology [15,16].

In addition to EVs, tumor-derived explants have emerged as another powerful tool in the fight against GBM. These 3D cell cultures meticulously recreate the architecture and behavior of the original tumors, capturing the intricate heterogeneity of cell populations within the tumor itself [17,18,19]. This detailed information provides a crucial roadmap for designing effective therapies that can comprehensively target all tumor subpopulations, ultimately improving patient outcomes [17,20,21,22]. Significantly, explants also accurately represent the tumor microenvironment (TME), the complex ecosystem that surrounds the tumor and plays a crucial role in its growth and spread. By studying explants, researchers can explore the intricate interactions between different cell types and signaling pathways within the TME, paving the way for the development of therapies targeting not only the tumor cells themselves but also the TME [22].

This study establishes a patient-derived GBM tumor culture system specifically designed to collect EVs released by the various cell types comprising the GBM TME in vitro, preserving its integrity and cellular architecture. Our primary objective is to identify markers specific to EVs within this cultured GBM TME, which could potentially enhance the future isolation of circulating tumor-associated EVs in patients. These markers hold promise for improving GBM detection methods and advancing therapeutic strategies. Additionally, investigating the molecular characteristics of circulating EVs derived from the GBM TME may yield critical insights into the features and progression of GBM.

## 2. Materials and Methods

### 2.1. Patient Samples

Nine GBM tissues were collected from patients undergoing surgical resection at the Department of Neurosurgery, Ospedali Riuniti di Livorno—USL Toscana Nord Ovest, Italy. This study was conducted according to the guidelines of the Declaration of Helsinki and approved by the Ethics Committee of the University Hospital of Pisa (787/2015). All patients provided written informed consent under a protocol approved by the Ethics Committee. All procedures were performed in accordance with relevant ethical guidelines and regulations.

Patient demographics are summarized in Table 1. Briefly, the group included 5 males (56%) and 4 females (44%) with a mean age of 73 years (range 66–80 years). All patients were diagnosed with GBM according to the WHO 2021 diagnostic criteria [23]. Importantly, tumors were selected for the absence of mutations in IDH1 and IDH2 genes (wild-type) and for the presence of 1p-19q co-deletion.

### 2.2. Tissue Processing and Storage

Following surgical resection, the neurosurgeon selected tumor specimens. A single pathologist examined all samples to ensure consistency. The resected tissues were collected and immediately stored in MACS tissue storage solution (Miltenyi Biotec, Bergisch Gladbach, Germany) at 4 °C for 2–4 h. Subsequently, tissues were viability-frozen at −140 °C in a solution containing 90% fetal bovine serum (FBS) (Thermo Fisher Scientific, Waltham, MA, USA) and 1% dimethyl sulfoxide (DMSO) (Sigma Aldrich, St. Louis, MO, USA). For each tumor fragment, cellularity and viability were assessed before storage by performing a tissue imprint on a slide, highlighted through immunohistochemical analysis, ensuring that only samples with high cellularity were selected for the study. All patient-derived surgical GBM tissues were de-identified before further processing.

### 2.3. GBM Explant Generation

Frozen tumor tissues were thawed rapidly in a 37 °C water bath. Subsequently, tissues were mechanically dissociated into a single-cell suspension following a protocol adapted from the study by Morelli et al. [21]. Tissues were washed with Dulbecco’s Phosphate-Buffered Saline (DPBS) in a sterile dish. Using a sterile scalpel, the tissues were then cut into small fragments (approximately 0.5 mm). The tumor fragments were collected in a 15 mL Falcon tube and washed three times with 10 mL aliquots of DPBS. After each wash, the solution was allowed to settle, and the supernatant was carefully aspirated to avoid losing the fragments. Following washes, the fragments were further processed through stacked pluriSelect cell strainer pluriStrainer (pluriSelect Life Science UG, Germany) to obtain a uniform size distribution and exclude very small or large pieces. The isolated fragments were then resuspended in a complete culture medium consisting of 50% DMEM/F12 medium (Thermo Fisher Scientific), 50% Neurobasal medium (Thermo Fisher Scientific), MEM-NEAAs solution (Thermo Fisher Scientific), GlutaMAX supplement (Thermo Fisher Scientific), penicillin-streptomycin (Thermo Fisher Scientific), N2 supplement (Thermo Fisher Scientific), B27 minus vitamin A supplement (Thermo Fisher Scientific), human recombinant insulin (Sigma Aldrich), and 2-mercaptoethanol (Sigma Aldrich), as previously described by Jacob et al. [19]. The final cell suspension was plated in 6-well multiwell plates and incubated on an orbital shaker rotating at 100 rpm within a 37 °C, 5% CO_2_, 90% relative humidity sterile incubator for 3 days (72 h).

### 2.4. Explant Immunohistochemistry (IHC)

Following explant collection, washing, and paraffin embedding processing, 5 μm thick tissue sections were placed on glass slides. The sections were then deparaffinized using xylene and rehydrated through a graded ethanol series.

IHC was performed using the Mouse and Rabbit Specific HRP/DAB (ABC) Detection IHC kit (ab64264, Abcam, Cambridge, UK) according to the manufacturer’s instructions. Antigen retrieval was achieved by microwave treatment with MS-unmasker solution (DIAPATH, Martinengo, BG, Italy). The following primary antibodies were used: GFAP mouse monoclonal antibody (ASTRO 6) (MA5-12023, Thermo Fisher Scientific) at 1:200 dilution, Ki67 Rabbit monoclonal antibody (SP6) (MA5-14520, Thermo Fisher Scientific) at 1:100 dilution, Iba1 Rabbit polyclonal antibody (019-19741, FUJIFILM Wako Chemicals, Richmond, VA, USA). Slides were developed with diaminobenzidine (DAB) chromogen (DAKO, Glostrup, DK) and counterstained with hematoxylin. Negative controls were generated by omitting the primary antibody incubation step. Images were captured using Aperio ImageScope viewing software version 12.4.6.5003 (Leica Biosystems, Nussloch, Germany).

### 2.5. EVs Collection and Purification

EVs were isolated from the cell culture supernatant using a two-step approach. First, cell debris was removed by differential centrifugation. The supernatant was collected and centrifuged at 3000× *g* for 15 min to pellet cellular debris. Next, the clarified supernatant was passed through a 0.22 μm filter (Millipore, Burlington, MA, USA) to eliminate larger vesicles. EVs enrichment was then performed using a commercially available membrane affinity-based isolation kit (exoEasy Maxi kit, Qiagen, Hilden, Germany) according to the manufacturer’s instructions.

### 2.6. Total Protein Quantification and Western Blot Analysis

For protein quantification, 5 µL aliquots of each sample were combined with 250 µL of Bradford reagent (Sigma Aldrich) in a 96-well plate. The mixture was incubated on a shaker at room temperature for 5 min, followed by absorbance measurement at 595 nm using a multiwell plate reader (Tecan, Mannedorf, Switzerland). A standard curve generated using bovine serum albumin (BSA) was used to determine the total protein concentration in each sample. All samples and standards were analyzed in triplicate.

Western blot analysis was performed to assess the presence of exosomal markers, CD63 and β-tubulin. Briefly, 5 μg of protein from each sample was loaded onto a 10% Mini-PROTEAN TGX gel (Bio-Rad, Hercules, CA, USA). Proteins were then transferred to a membrane using the Trans-Blot Turbo transfer system (Bio-Rad). After the blocking step with 5% milk (Sigma Aldrich), the blots were incubated with primary antibodies: CD63 (1:250 dilution, #10628D, Thermo Fisher Scientific) and β-tubulin (1:250 dilution, #HPA043640, Sigma Aldrich). After washing, the blots were incubated with horseradish peroxidase (HRP)-conjugated secondary antibodies: Goat AntiMouse IgG H&L (HRP) (1:2000 dilution, #ab6789, Abcam) and Goat AntiRabbit IgG H&L (HRP) (1:2000 dilution, #ab6721, Abcam). Protein bands were visualized using Bio-Rad Clarity Western ECL substrate (Bio-Rad) and imaged with a ChemiDoc MP imager using Image Lab software version 6.0.1 (Bio-Rad).

### 2.7. Multiplex Analysis of EVs Surface Markers

Surface protein expression of isolated EVs was analyzed using the Human MACSPlex Exosome Kit (130-108-813, Miltenyi Biotec, Bergisch-Gladbach, Germany) according to the manufacturer’s instructions for the overnight protocol with the MACSPlex Filter Plate. Briefly, 15 μg of protein from each EV sample was diluted to a final volume of 120 μL using MACSPlex Buffer. A total of 15 μL of Capture Beads was then added to each well of the filter plate, and the plate was incubated overnight at room temperature on an orbital shaker (450 rpm) protected from light. After washing with MACSPlex Buffer, a cocktail of Detection Reagents specific for CD9, CD63, and CD81 was added to each well. The plate was again incubated for 1 h at room temperature on an orbital shaker (450 rpm) protected from light. Following additional washes, samples were analyzed using the CytExpert Software for the CytoFLEX S Platform (Beckman Coulter, Brea, CA, USA). Following the detection of Capture Bead populations in a FITCH versus PE dot plot, each population was assigned the corresponding target marker (CD9, CD63, or CD81). The relative levels of EV surface markers were determined as follows. The median signal intensity of each microsphere from the control sample (buffer only) was subtracted from the signal intensity of the respective microspheres incubated with the EV sample. The median signal intensity for each sample was then calculated for the MACSPlex CD9, CD63, and CD81 Capture Beads. Finally, the median signal intensity of the MACSPlex EV Capture Beads for CD9, CD63, and CD81 was used as a normalization factor for each sample. This normalizes the data by setting the mean signal intensity of the Capture Beads to a constant value.

### 2.8. RNA Expression Analysis in GBM and Normal Controls

Publicly available online databases were utilized to analyze the gene expression of EV markers in GBM and healthy controls. The Gene Expression Profiling Interactive Analysis (GEPIA) tool [24] was employed to investigate the expression of these markers in 163 GBM cases and 207 healthy controls (http://gepia2.cancer-pku.cn/, accessed on 29 January 2024). Differential expression analysis was performed to identify genes with significant expression differences between tumors and healthy controls, both for individual markers and for all markers considered collectively. GEPIA further generated a principal component analysis (PCA) visualized as a 3D plot of the first three principal components and a curve analysis for overall survival. For a more comprehensive analysis, Pearson correlation analyses were conducted on a larger cohort of 528 GBM samples from The Cancer Genome Atlas (TCGA) GBM dataset (HG-UG133A) accessed through the GlioVis portal (https://gliovis.bioinfo.cnio.es/, accessed on 7 February 2024) [25]. This portal was also used to explore spatial RNA-sequencing (RNA-seq) data from the Ivy Glioblastoma Atlas Project (IVY GAP) [26]. This dataset encompasses 122 RNA samples from 10 patients and provides expression data for five distinct anatomical structures within the tumor identified by hematoxylin and eosin (H&E) staining: leading edge, infiltrating tumor, cellular tumor, microvascular proliferation, and pseudopalisadic cells around necrosis. Finally, the GBMseq portal (http://gbmseq.org/, accessed on 9 February 2024) was employed to examine the expression of selected EVs membrane markers in major cell types of the central nervous system (CNS) using single-cell RNA-sequencing (scRNA-seq) data. This analysis focused on vascular, immune, neuronal, and glial cells.

## 3. Results

Our study included nine patients with GBM who underwent total resection and did not present with IDH1, IDH2 mutations, or 1p-19q co-deletion. Patient characteristics and tumor details are provided in Table 1. The cohort comprised five men (56%) and four women (44%), with an average age of 73 years (±5). This selection process aimed to ensure a relatively homogeneous group of GBM cases for further analysis, taking into account not only tumor characteristics but also the sex and age distribution of patients.

### 3.1. Establishment of a Tumor Culture System for Efficient Production and Isolation of EVs

We developed a protocol to generate explants from the tissues of GBM patients to collect secreted EVs (Figure 1A). This method prioritizes minimal manipulation and rapid processing. The tumor sample is cut into small, uniform pieces with an approximate diameter of 0.5 mm. To achieve a consistent fragment size, the sample undergoes a series of filtration and washing steps. Subsequently, the prepared tissue pieces are cultured in suspension within shake flasks, which facilitates enhanced aeration and efficient exchange of nutrients and metabolic waste products in the surrounding medium.

To assess the model’s efficacy in preserving TME features, we evaluated cell viability and tumor architecture after one week of culture (Figure 1B). Both parameters were well-maintained, indicating the model’s ability to support tissue integrity for a week. Hematoxylin and eosin (H&E) staining revealed typical GBM cellular and tissue structures (Figure 1B). Ki67 staining, a marker for proliferating cells, showed active proliferation throughout the tissue. Additionally, GFAP and IBA1 staining visualized astrocytes and tumor-associated microglia/macrophages (TAMs), respectively, with a distribution mirroring real GBM tissue. These findings demonstrate the model’s effectiveness in preserving the complex TME.

Next, we isolated EVs from explants cultured for 72 h. Our multistep isolation protocol involved differential centrifugation, filtration, and a final membrane affinity binding step. We based our approach on previous validations of this membrane-based method, such as the study by Enderle et al. [27], which demonstrated that this technique allows for the efficient recovery of intact, morphologically regular EVs while avoiding many of the contaminants. This approach yielded an average protein concentration of 0.52 µg/µL in the isolated EVs (Appendix A). Western blot analysis confirmed the presence and relative abundance of exosomes in all samples through detection of the exosomal marker CD63 (Figure 1C). The strong and consistent expression of CD63 across all EV samples derived from the cultured tumor model suggests the presence of exosomes.

### 3.2. Characterization of EVs Surface Protein Composition Using Multiplex Protein Analysis

We employed the MACSPlex multiplex assay to evaluate the surface protein expression profiles of tumor-derived EVs produced in our GBM culture model. This assay facilitates the simultaneous analysis of 37 surface epitopes, along with two isotype controls, serving as a measure of nonspecific binding to determine background signal levels [28,29]. After quantification and dilution, the EV preparations were adjusted to a final protein concentration of 15 µg, as specified in the data sheet. The multiplex platform utilizes polystyrene beads with a diameter of 4.8 mm, labeled with varying amounts of two dyes to generate 39 distinct bead populations distinguishable by flow cytometry (Figure 2A, left). Each bead population is conjugated to a unique capture antibody that specifically binds to EVs carrying the corresponding antigen. EVs bound to the beads are then detected using a cocktail of exosomal marker antibodies, including anti-CD9-APC, anti-CD63-APC, and anti-CD81-APC (Figure 2A, right).

As illustrated in Figure 2B, both negative controls (mIgG1 and REA controls) showed low signal intensities, with average signals between 0.4 and 0.5. These results validate the specificity of the assay, confirming that the observed signals are attributable to EVs binding rather than nonspecific interactions with other cellular components. The mean fluorescence intensity (MFI) of 24 out of the 37 markers analyzed (including CD1c, CD2, CD3, CD4, CD8, CD11c, CD19, CD20, CD24, CD25, CD31, CD40, CD41b, CD42a, CD45, CD49e, CD62P, CD69, CD86, CD209, CD326, SSEA-4, ROR1, and HLA-ABC) was similar to or slightly above the MFI of isotype controls, suggesting that these markers exhibit low expression (below the detection limit of the assay) or are absent from the EVs produced by the cultured tumors in all nine samples analyzed (Figure 2B). Conversely, the remaining 13 markers (CD105, CD133/1, CD14, CD142, CD146, CD29, CD44, CD56, HLA-DR/DP/DQ, MCSP, and the three tetraspanins CD9, CD63, and CD81 specially enriched in the membrane of EVs and often used as exosome biomarkers) showed higher MFIs than their respective isotypic controls, indicating their presence on the surface of the tumor-derived EVs (Figure 2B,C). Details of the expression levels of the selected genes within each individual sample are shown in Appendix A. This kit uses tetraspanins CD9, CD63, and CD81 as established EV-associated markers to capture and detect the total EV population. The assay employs these tetraspanins as detection reagents, facilitating the binding of EVs to MACSPlex Capture Beads and enabling subsequent analysis of additional markers. Through this approach, sandwich complexes are formed between the EVs, capture beads, and detection reagents, allowing for phenotypic analysis of the captured EVs based on fluorescence characteristics. The expression patterns of these markers in our samples revealed that CD81 appears to be the most robust indicator of EVs produced by GBM cells (Figure 2D).

The presence of EV markers associated with various cell types underscores the heterogeneous nature of GBM tumors in culture. Table 2 provides a detailed account of how the identified EV markers have been described so far in the literature on GBM tumors. Notably, these markers have previously been linked to different cell types, including endothelial cells (CD44, ENG, F3, MCAM, and ITGB1), immune cells (CD14, CD44, CSPG4, ENG, HLA-DR/DP/DQ), and neuronal cells (NCAM1), as well as astrocytes and tumor cells. This observation lends further credibility to our in vitro model, suggesting that it effectively maintains the dynamic interplay of cells constituting the TME, thus preserving their activity and vitality.

### 3.3. Tumor-Derived EV Membrane Biomarker Expression in GBM

To further investigate the biomarkers identified on the membranes of EVs produced by tumors in culture, we explored their expression patterns in GBM. It is important to note that the HLA-DR/DP/DQ marker was not included in the analysis because of its complexity, as it encompasses multiple loci, including HLA-DR, -DQ, and -DP, each composed of several subunits. We analyzed a case series of 163 RNA-Seq data files retrieved from The Cancer Genome Atlas (TCGA) database and compared them with 207 healthy control samples of human cerebral cortex obtained from the Genotype-Tissue Expression (GTEx) project using the Gene Expression Profiling Interactive Analysis (GEPIA) tool [24]. Our objective was to investigate variations in the expression levels of the nine selected EV membrane markers across the series of samples. As demonstrated in Figure 3A, all biomarkers exhibited higher expression levels in GBM samples (T) compared with healthy controls (N).

We next conducted a multivariable analysis to better understand the collective behavior of the EV markers. The results, displayed in Figure 3B, reveal that the simultaneous expression of all EV markers was significantly higher (*p* < 0.05) in GBM samples than in healthy controls. A differential gene expression analysis was also performed for individual genes (Appendix A), revealing that eight out of nine genes were significantly overexpressed in the 163 TCGA GBM samples compared with healthy GTEx controls. This collective increase in biomarker expression in GBM suggested the potential of these markers as a collective signature of the disease. To understand even better the differences in the expression of these nine EV biomarkers between tumor and control brain tissues, we performed a principal component analysis (PCA). The resulting visualization, shown in Figure 3C, presents a clear distinction between the two groups of samples. The blue cloud representing GBM tumor samples and the green cloud representing healthy controls were significantly separated, highlighting the unique gene expression profiles of EVs biomarkers in the two tissue types. Finally, we wanted to determine whether there was a correlation between the expression of these nine EV biomarkers and the survival rates of the 163 patients diagnosed with GBM in the TCGA database. Kaplan-Meier curves were generated for individual genes (Appendix A). Of the nine genes analyzed, only three—CD14, MCAM, and CD44—were significantly overexpressed in the group with worse survival outcomes.

### 3.4. Correlation of Tumor-Derived EV Membrane Biomarker Expression

Figure 4 shows the Pearson correlation analyses between the gene expressions of the nine selected biomarkers using a cohort of 528 GBM samples from the TCGA GBM dataset (HG-UG133A) through the GlioVis portal [25]. Each box on the left side of the matrix represents the distribution of a specific biomarker, while the right side shows the Pearson correlation coefficients and corresponding significance levels. A closer examination of the matrix reveals a network of statistically significant (*p* < 0.05) correlations within distinct clusters, suggesting potential co-expression patterns among the selected biomarkers.

A noteworthy observation is the strong positive correlation between ENG (endoglin) and CD14, indicating that these markers tend to be co-expressed in GBM samples. In addition, a moderate positive correlation is observed between ENG and MCAM (Melanoma Cell Adhesion Molecule), as well as between CD14 and CD44. Other weaker but still significant correlations exist between ENG with F3 (Tissue Factor) and CD44, between PROM1 (Prominin-1) with CSPG4 (Chondroitin Sulfate Proteoglycan 4) and NCAM1 (Neural Cell Adhesion Molecule 1), between CD14 with F3 and MCAM, between F3 with CSPG4 and MCAM, between MCAM and NCAM1, and between NCAM1 and CSGP4.

Some slight but significant negative correlations are also observed between some genes. These include ENG and PROM1, PROM1 with CD44 and ITGB1 (Integrin Subunit Beta 1), CD14 and CSPG4, F3 with ITGB1, ITGB1 with NCAM1, and CD44 with NCAM1 and CSPG4.

### 3.5. Expression of EV Markers in Spatial and Single-Cell RNA-Seq Analysis

The TME is composed of a diverse array of cell populations. Bulk RNA-sequencing (RNA-seq) often fails to adequately capture this cellular heterogeneity and the complex interactions between these cell types. To address this limitation, we investigated the expression of EV markers in both histologically distinct regions of the tumor and in the various cell types that constitute the TME. We utilized spatial RNA-seq data from the Ivy Glioblastoma Atlas Project (IVY GAP) database accessed through the GlioVis portal [25]. This database encompasses 122 RNA samples from 10 patients and provides expression data for five distinct anatomical structures within the tumor identified by H&E staining: leading edge, infiltrating tumor, cellular tumor, microvascular proliferation, and pseudopalisading cells around necrosis. Additionally, we analyzed single-cell RNA-seq (scRNA-seq) data through the GBMseq portal (http://gbmseq.org/, accessed on 9 February 2024) to examine the expression of selected EVs membrane markers across major central nervous system cell types, including vascular, immune, neuronal, and glial cells. The expression patterns of these EV markers within the various histological zones of GBM are illustrated in Figure 5A. Notably, CSPG4, ENG, ITGB1, and MCAM displayed high expression within the microvascular proliferation zone. Conversely, CD44 and PROM1 exhibited particularly high expression in the pseudopalisades surrounding necrotic cells. CD14, F3, and NCAM1 did not demonstrate exclusive expression within any specific histological zone. As depicted in Figure 5B, certain markers exhibited cell type-specific expression patterns. For instance, CD14 displayed predominant expression in myeloid-derived cells, particularly macrophages and microglia. Conversely, CSPG4 and PROM1 showed predominant expression in oligodendrocyte precursor cells (OPCs). Endothelial cells demonstrated the highest expression of ENG, ITGB1, and MCAM compared with other cell types. Among neoplastic cells, CD44 emerged as the most highly expressed marker. Notably, NCAM1 displayed high expression across several cell types, including neurons, oligodendrocytes, OPCs, astrocytes, and neoplastic cells, while endothelial and myeloid cells exhibited lower expression levels.

## 4. Discussion

This study presents a valuable platform for isolating, identifying, and characterizing tumor-derived EVs using a novel in vitro explant model derived from patient-specific GBM tissue. Unlike traditional cell lines or explants obtained through complete tumor dissociation, our model preserves the cellular heterogeneity and complex interactions intrinsic to the GBM tumor microenvironment (TME).

The importance of developing 3D models that closely mimic the TME lies in their ability to provide a more accurate reflection of the disease. Traditional 2D models, while easy to use, fail to capture the intricate spatial and cellular interactions present in the TME, limiting understanding of tumor behavior and therapeutic response [48,49]. In contrast, 3D models, such as tumor-derived explants, more effectively replicate the architecture and heterogeneity of the original tumor [50]. These models allow the study of critical cell–cell and tumor–normal tissue interactions, which are essential for understanding tumor progression and mechanisms of drug resistance [51].

EV analysis has emerged as a powerful tool for unraveling the complexities of cell–cell communication within the TME, particularly in GBM. These extracellular vesicles play a fundamental role in horizontal information transfer, facilitating the exchange of proteins, lipids, and nucleic acids between cells [10,52]. This communication pathway is particularly critical in cancer, where EVs contribute to tumor progression and therapeutic resistance [53]. Our GBM culture model offers a unique advantage by enabling the collection of EVs produced by the entire TME. This comprehensive approach allows for a deeper understanding of the various cell types involved in GBM progression. By identifying GBM-specific EV markers, we aim to enhance the specificity of tumor-derived EV isolation from patient plasma. This could potentially improve the reliability of EVs-based assays for disease monitoring and contribute to the development of targeted therapeutic strategies in GBM. The multiplex kit employed in this study enables the simultaneous quantification of 37 different membrane proteins, providing valuable insights into the molecular composition of these EVs. The characterization of EV membrane proteins is particularly important, as they are essential for EV biogenesis, targeting, and uptake by recipient cells [54,55]. Furthermore, the identification of specific membrane protein signatures could serve as potential biomarkers for early diagnosis, prognosis, and the development of personalized therapeutic strategies [56,57,58].

Nine GBM samples with homogeneous characteristics were chosen. All selected tumors were primary GBMs, lacking IDH mutations or 1p19q chromosomal codeletion. Additionally, patient demographics were carefully controlled, with a narrow age range and equal sex distribution to minimize potential confounding variables. A multiplex protein analysis was employed to identify 37 distinct membrane epitopes frequently expressed on EVs. This comprehensive approach facilitated the selection of 13 consistently expressed membrane proteins across all samples. Notably, these 13 markers included the well-established exosomal membrane markers: tetraspanins CD9, CD63, and CD81 [59]. Interestingly, CD81 emerged as the most abundantly expressed tetraspanin among the identified markers in our GBM EVs. This observation aligns with existing literature that associates high CD81 expression with cancerous tissues and cell lines [60,61]. Furthermore, recent research suggests a pivotal role for CD81 in DNA repair via regulation of Rad51 nuclear transport, potentially positioning it as a promising therapeutic target for GBM radiotherapy [62]. In addition to tetraspanins, our multiplex analysis identified several other proteins consistently expressed on the EVs membranes isolated from all cultured GBMs. These proteins included Englin (ENG or CD105), Prominin-1 (PROM1 or CD133/1), CD14, Tissue Factor (F3 or Coagulation Factor III or CD142), Melanoma Cell Adhesion Molecule (MCAM or CD146), Integrin Subunit Beta 1 (ITGB1 or CD29), CD44, Neural Cell Adhesion Molecule 1 (NCAM1 or CD56), Major Histocompatibility Complex Class II isotypes HLA-DR, -DQ, and -D (HLA-DR/DP/DQ), and Chondroitin Sulfate Proteoglycan 4 (CSPG4 or MCSP). Our findings regarding the expression of EV markers within the TME align with established knowledge of their functions in various cell types.

CD105 (endoglin, ENG), a TGFβ-binding protein highly expressed in tumors undergoing active vascularization and highly expressed in zones of microvascular proliferation, is associated with angiogenesis and poor prognosis in GBM [63,64]. It has also been found on microvesicles, promoting a pro-angiogenic and pro-metastatic microenvironment [65]. Similarly, CD133 (PROM1), a stem cell marker found particularly enriched in pseudopalisades surrounding necrosis, correlates with tumor growth, therapy resistance, and recurrence in GBM stem cells, with high levels observed in EVs from both melanoma and glioma [32,66].

CD14, typically involved in innate immune responses and predominantly expressed in myeloid-derived cells, is linked to immunosuppression and tumor invasiveness, with CD14+ exosomes promoting metastasis in cancer [67,68]. Tissue Factor (TF, CD142), expressed in extravascular cells (fibroblasts, smooth muscle), is implicated in tumor progression, therapy resistance, and enhanced pro-angiogenic signaling, with TF+ EVs fostering a pro-tumor microenvironment [69,70]. CD146 (MCAM), expressed in endothelial cells, pericytes, and mesenchymal cells, is associated with mesenchymal features and poor prognosis in GBM, and its presence on EVs has been linked to organ targeting and metastatic niche formation [71,72].

Integrin beta-1 (ITGB1, CD29) and CD44, with their widespread expression in various cell types, are both key mediators of tumor invasion, metastasis, and resistance in GBM and other cancers. ITGB1+ EVs are involved in ECM interactions and metastatic niche formation [73,74], while CD44+ EVs are linked to tumor recurrence and therapy resistance in GBM, suggesting their role as biomarkers for aggressive tumor subtypes [42,75]. CD56 (NCAM1), expressed in neural stem cells and natural killer cells, is similarly overexpressed in GBM, contributing to chemoresistance and poor prognosis, and is found on exosomes, where it may modulate immune responses [45,76,77].

HLA-DR, HLA-DP, and HLA-DQ, although associated with poor prognosis in GBM [78], were not investigated in detail due to the complexity of their subtypes, despite their known involvement in immune modulation via exosomes [79]. CSPG4 (MCSP), expressed in mesenchymal cells, is linked to stem cell renewal, tumor recurrence, and drug resistance and plays a key role in GBM progression through interactions with the ECM and angiogenesis, also contributing to multidrug resistance via PI3K pathway activation [80,81]. These findings reinforce the importance of EV-associated markers in GBM progression and their potential as therapeutic targets or biomarkers for disease monitoring.

Given the expression of CD9, CD63, and CD81, which are well-established markers associated with exosomes, along with the presence of additional membrane proteins previously described in the literature as exosomal markers, our findings suggest the presence of exosomes within the analyzed EVs. This underscores the relevance of these markers in identifying the exosomal subpopulation and highlights the need for further investigation into their functional implications within the TME.

The presence of EVs expressing markers from diverse cell types within the TME underscores the complexity and biological fidelity of our cultured tumor model. This in vitro system appears to preserve the viability and functionality of various cell subpopulations. Furthermore, the entire protocol encompassing culture, EV isolation, and analysis could be adapted for further investigations. This includes exploring the role of EVs in cell communication and cell–cell interactions within the TME.

This study next analyzed the gene expression of markers found in online datasets of GBM tumors and healthy controls. Compared with normal tissue, GBM samples showed significant overexpression of these markers. In particular, for some genes, namely CD14, MCAM, and CD44, even higher expression levels were observed in patients with worse survival outcomes. Interestingly, the expression levels of these markers showed both positive and negative correlations, suggesting potential regulatory mechanisms within the TME. Given the unavailability of EV datasets from healthy brain cells, for direct comparison with GBM-derived EVs, we chose instead to analyze mRNA expression data from healthy versus tumor tissue. This approach offered insights into the potential relevance of these markers in distinguishing tumor tissue from healthy controls and enabled us to examine their expression across different cell types and histological regions associated with GBM pathology. To further study intratumoral heterogeneity, we analyzed the expression of these markers in different GBM cell subtypes, again making use of online databases. Our analyses confirmed the differential expression of these markers in distinct histological regions of the tumor. Next, we employed single-cell analysis to further investigate cellular heterogeneity and evaluate the expression of the markers in individual cell types. This approach confirmed the specific association of certain markers with distinct cell populations within the GBM TME. These results highlight the presence of histologically distinct areas and different cell subtypes within our cultured tumors, reflecting the nature of the original tumor.

Future efforts should focus on expanding the patient cohort to validate these findings across a larger population. A more comprehensive case series would allow for a deeper understanding of the variability and specificity of the EV markers identified in GBM. Additionally, the analysis of circulating EVs from the plasma of the same patients could provide further insights into whether these EV signatures are present in the bloodstream, thereby offering a minimally invasive method for monitoring tumor dynamics. To enhance the specificity of these markers, it will be crucial to compare EV data from GBM patients with that from healthy controls. This will help determine whether the identified markers are exclusive to GBM or are more broadly associated with other conditions. Furthermore, functional studies should be undertaken to investigate the roles these EVs play in tumor progression, immune modulation, and drug resistance. Understanding their biological functions could pave the way for their use as prognostic indicators or therapeutic targets. Finally, integrating EV profiles with multiomics data, such as DNA and RNA analysis, would provide a more comprehensive view of GBM biology. This approach could reveal novel insights into the molecular pathways involved in GBM and uncover new opportunities for targeted therapies.

## 5. Conclusions

In conclusion, this study successfully isolated and characterized EVs produced by different cell populations within GBM tumors, offering a valuable tool for studying intercellular communication in GBM. Furthermore, the development of a 3D in vitro explant model derived from GBM patient samples enabled this innovative approach, enhancing our understanding of the tumor microenvironment. One key achievement of this study is the identification of shared EV surface markers across all analyzed tumor samples. This finding suggests the presence of EV markers that are potentially specific to GBM, which could be valuable for diagnostic and therapeutic purposes. While the limited number of cases in this study precludes broad generalizations, it may represent the first characterization of EV surface markers produced by human GBM explants, highlighting the potential of EV-based biomarkers in GBM research.

## Figures and Tables

**Figure 1 cancers-16-03748-f001:**
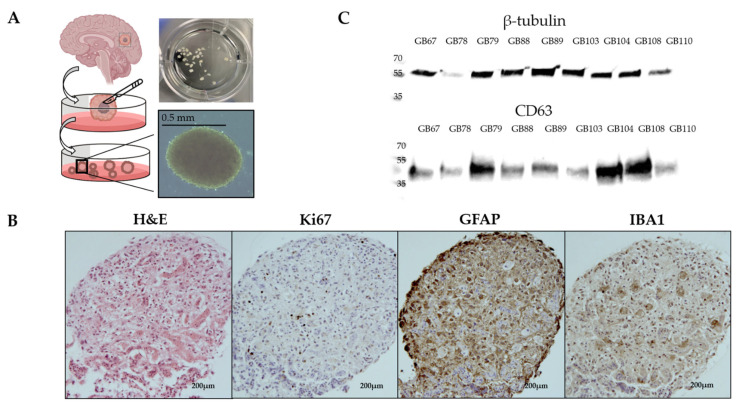
Isolation and characterization of EVs produced by patient-derived tumors: (**A**) Outline of our experimental protocol illustrating the cultivation of patient-derived explants, physiological release of EVs into the culture medium, and subsequent isolation and characterization of these EVs for further investigation. (**B**) Immunohistochemical staining of a tumor-derived explant section after one week of culture. H&E, hematoxylin and eosin staining; Ki67, nuclear marker for proliferating cells; GFAP, astrocyte glial fibrillary acidic protein staining; IBA1, ionized calcium-binding adapter molecule 1 staining of tumor-associated microglia/macrophages. (**C**) Western blot analysis of exosomal marker CD63 and β-tubulin expression in EVs isolated from patient-derived tumor culture medium.

**Figure 2 cancers-16-03748-f002:**
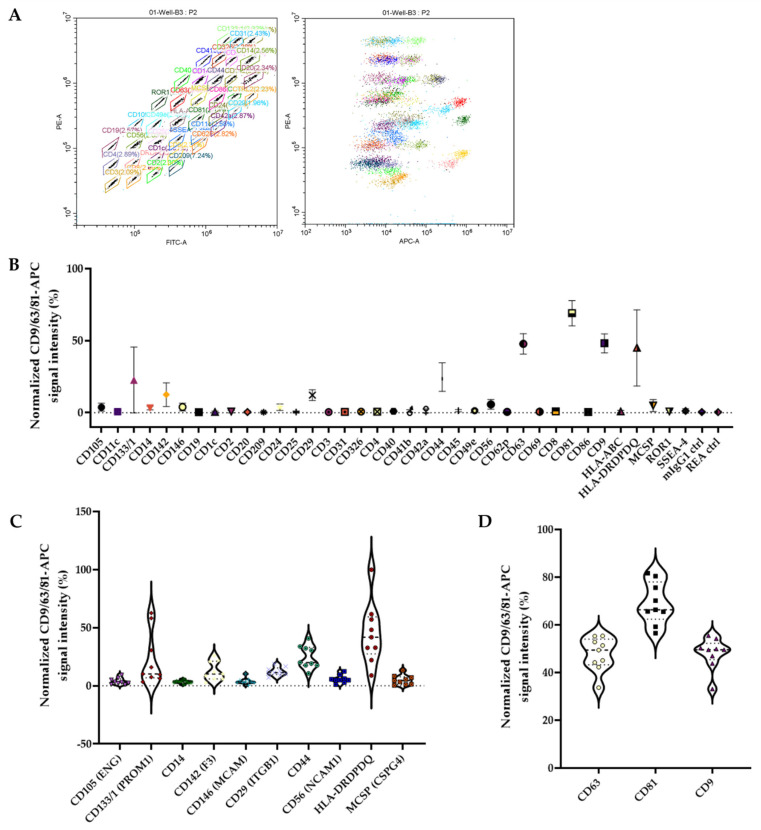
Flow cytometry analysis of EVs membrane markers using the MACSplex kit: (**A**) The left panel illustrates the gating strategy employed to differentiate between distinct bead populations labeled with various markers. The right panel displays the signal intensity measurements of individual bead populations, enabling the characterization of EV membrane markers. (**B**) Overview of the total expression profile for 39 surface markers on EVs isolated from GBM cultures. Data are presented as normalized and background-subtracted median CD9/CD63/CD81 fluorescence intensity (MFI) (isotype control and blank samples). (**C**) Expression profile of the 10 proteins of the 37 analyzed that showed positive signals (higher MFIs compared with the respective isotype controls) in all tumor samples. (**D**) Expression profile of the three tetraspanins (CD9, CD63, and CD81) present on the surface of the GBM-derived EVs under investigation.

**Figure 3 cancers-16-03748-f003:**
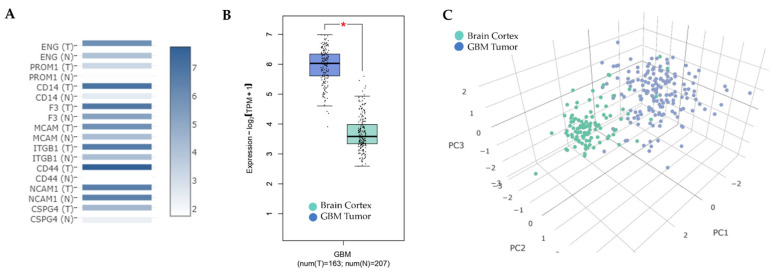
EVs Membrane Biomarker Expression in GBM: (**A**) Expression matrix of EVs markers, with color density representing median expression values of genes in GBM (T) and control human cortex samples (N), normalized to the maximum median expression value. Log2(TPM + 1) was used for log scaling. (**B**) Box plots with jitter comparing expression in GBM samples (blue box) expression in comparison to control tissue (green box). Log2FC cutoff: 1 and *p*-value cutoff: 0.01 were used for differential thresholds. One-way ANOVA was employed for differential analysis. (**C**) Principal component analysis of the nine EVs markers based on GBM (blue dots) and controls (green dots) expression data, presented in a 3D plot of the first three principal components. Analyses were conducted using the GEPIA2 tool, which utilizes data from The Cancer Genome Atlas (TCGA) for the expression analysis of 163 GBM samples and the Genotype-Tissue Expression (GTEx) database for comparison with the expression data from 207 control cortex samples.

**Figure 4 cancers-16-03748-f004:**
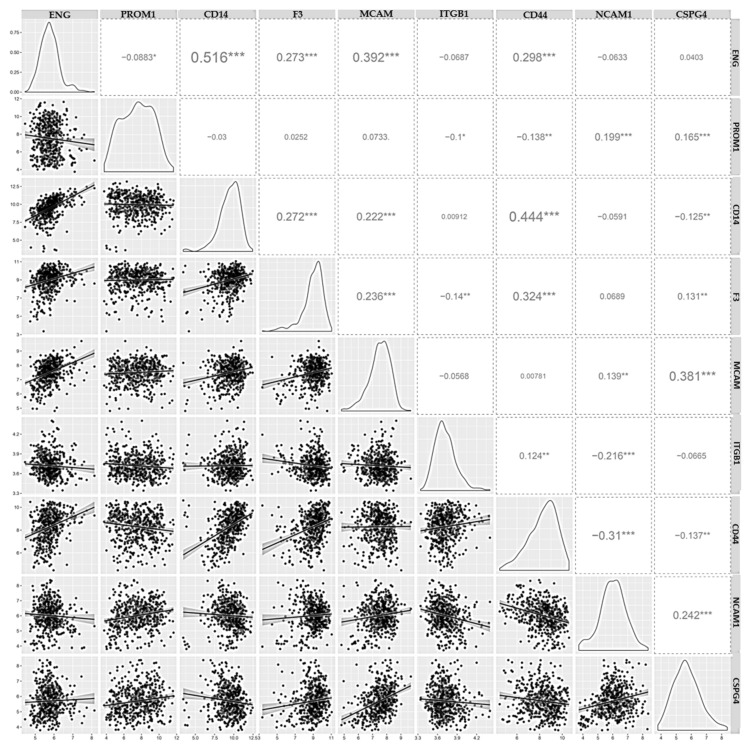
Correlation of biomarker expression. Correlation matrix of the expression of the 9 selected markers in 528 GBM samples. The distribution of variables is shown on the left, and Pearson correlation coefficients and significance level are shown on the right. X and Y labels correspond to the log-transformed values of the biomarkers. (*** *p* < 0.001; ** *p* < 0.01; * *p* < 0.05). Correlation analysis was conducted using the Gliovis portal, which utilizes data from The Cancer Genome Atlas (TCGA) for the expression analysis of 528 GBM samples.

**Figure 5 cancers-16-03748-f005:**
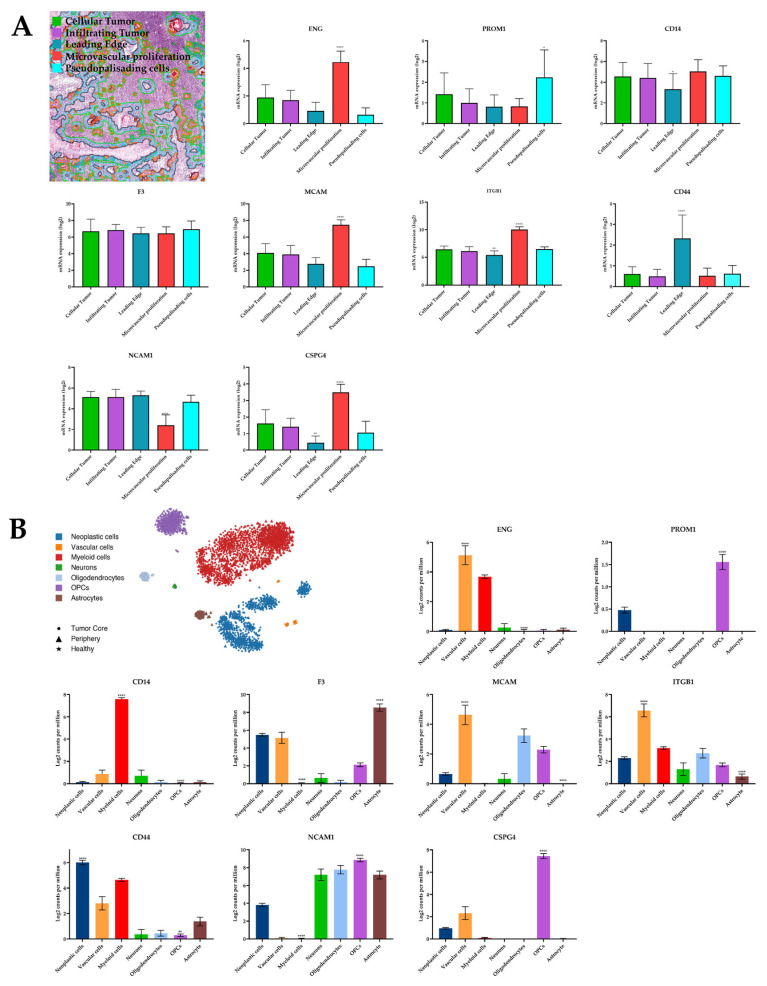
(**A**) Analysis of the nine EVs membrane marker expression in GBM tissue using RNA-seq of five isolated anatomical structures by laser microdissection: cellular tumor (red), infiltrating tumor (yellow), leading edge (green), microvascular proliferation (blue), and pseudopalisading cells around necrosis (purple). The analysis and graphical visualization of the data were conducted utilizing GlioVis portal. (**B**) Analysis of the expression of nine EVs membrane markers in GBM tissue using RNA-seq of single cells from seven distinct cell types, each representing a separate cluster: neoplastic cells (blue), vascular cells (orange), myeloid cells (red), neurons (green), oligodendrocytes (light blue), oligodendrocyte progenitor cells (OPCs, purple) and astrocytes (brown). The analysis and graphical visualization of the data were conducted utilizing the user-friendly web interface provided by the gbmseq.org platform. Statistically significant differences were highlighted only for the group displaying significant overexpression or downregulation compared with other groups for each gene, using an unpaired *t*-test. Significance levels are denoted as follows: * = *p* ≤ 0.05; ** = *p* ≤ 0.01; **** = *p* ≤ 0.0001.

**Table 1 cancers-16-03748-t001:** Patient and tumor characteristics. Abbreviations: WT = wild type; M = male; F = female.

	Tumor	Sex	Age	IDH1	IDH2	1p-19q
1	GB67	F	76	WT	WT	absent
2	GB78	M	71	WT	WT	absent
3	GB79	M	79	WT	WT	absent
4	GB88	M	70	WT	WT	absent
5	GB89	M	78	WT	WT	absent
6	GB103	F	69	WT	WT	absent
7	GB104	F	80	WT	WT	absent
8	GB108	M	66	WT	WT	absent
9	GB110	F	71	WT	WT	absent

**Table 2 cancers-16-03748-t002:** Proteins detected by multiplex bead-based flow in EVs produced by GBM-derived tumors.

Symbol	Name	Putative Cell Type	Known Function	Potential Tumor Relevance
CD105 (ENG) [30]	Endoglin	Endothelial cells, microglia	Cellular proliferation, differentiation, and migration	Regulation of angiogenesis, both under physiological and pathological conditions
CD133/1 (PROM1) [31,32,33]	Prominin 1	Stem cells, progenitor cells	Cell self-renewal, proliferation, differentiation	Contributes to tumor initiation, maintenance, and resistance to therapy.
CD14 [34,35]	Cluster of Differentiation 14	Macrophages, neutrophils, dendritic cells	Immune activation	Can promote tumor growth and invasion by suppressing immune response or acting as tumor-associated macrophages (TAMs).
CD142 (F3) [36,37,38]	Tissue Factor	Extravascular cells (fibroblasts, smooth muscle)	Angiogenesis, cell survival, cell proliferation	Regulates blood vessel formation and function, potentially influencing tumor growth and drug delivery.
CD146 (MCAM) [39,40]	Melanoma Cell Adhesion Molecule	Endothelial cells, pericytes, Mesenchymal cells	Vascular development, cell adhesion, inflammation, migration, invasion	May contribute to angiogenesis and blood-brain barrier disruption, aiding tumor progression.
CD29 (ITGB1) [41]	Integrin Subunit Beta 1	Various cell types, including epithelial, endothelial, immune and mesenchymal cells	Cell adhesion, migration, extracellular matrix interaction	Modulating cell–cell and cell–matrix interactions, potentially promoting tumor invasion and metastasis
CD44 [42,43,44]	CD44 Molecule	Mesenchymal stem cells, neural stem cells, myeloid cells, vascular cells	Cell adhesion, migration, signal transduction	Associated with cancer stem cell properties, tumor invasion, and resistance to therapy.
CD56 (NCAM1) [45]	Neural Cell Adhesion Molecule 1	Neural stem cells, natural killer cells	Cell adhesion, immune response	Controlling cell–cell interactions and adhesion and potentially promoting tumor progression and metastasis
HLA-DR/DP/DQ [46]	Major Histocompatibility Complex	Antigen-presenting cells (APCs)	Antigen presentation, T cell activation	Crucial for immune recognition and response against tumor cells, but dysfunction can lead to immune escape.
MCSP (CSPG4) [47]	Chondroitin Sulfate Proteoglycan 4	Mesenchymal cells	Cell adhesion, migration	May contribute to the migratory and adhesive properties of GBM cells within the microenvironment.

## Data Availability

This study did not generate any new publicly archived datasets. All data supporting the reported results are described within the manuscript and are derived from previously published sources as cited.

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
