# Peer review of "Exploring Extracellular Vesicle Surface Protein Markers Produced by Glioblastoma Tumors: A Characterization Study Using In Vitro 3D Patient-Derived Cultures"

_cancers, 2024, doi:10.3390/cancers16223748_

Round 1
Reviewer 1 Report
Comments and Suggestions for Authors
Exploring exosome surface protein markers produced by glioblastoma tumors: a characterization study using in vitro 3D patient-derived cultures
The manuscript by Franceschi S et.al., focuses on establishing a patient-derived GBM tumor culture system specifically designed to collect exosomes produced and released from all cell types that comprise the tumor, while maintaining the integrity of the tumor microenvironment and cytological architecture. Their main goal has been to identify GBM-specific markers on the exosomal surface to improve the isolation of circulating exosomes in patients. These biomarkers demonstrated potential as a GBM-specific signature and were correlated with clinical data and hold great promise for the development of new GBM detection methods and therapeutic strategies.
This manuscript is well described and have few minor experiments which I feel will strengthen the paper.
My major concern is the use of the affinity binding pull down of exosomes using the kit from Qiagen. Since the whole study is focussed on the proteins surface markers, there has been debate using kit that precipitates soluble proteins along with other protein aggregates with exosomes. This precipitation will give false positive data for studying the protein markers for GBM. The authors should consider using TFF and using SEC to enrich and purify exosomes/EVs from their patients tissue derived conditioned media. Based on the evidence provided by the authors in Figure 1A, they get sufficient material to work with tissue and isolating EVs should not be an issue. Once isolated, it would be ideal to compare the protein markers with the current precipitation method which the authors used in their study.
The authors also needs to be careful while using the term exosomes, The author doesn’t have enough evidence to show that the exosomes produced are from endocytic origin. They should provide evidence with TEM and NTA and couple of more markers with WBs to convince that the EVs isolated are indeed EVs and not just bound /degraded CD63 membrane proteins. The authors should nomenclate EVs through out the manuscript as no evidence is provided. Please follow the latest MISEV guidelines published this year 2024 and nomenclate accordingly.
Fig 2,3 and 5 needs to be made again with clear font and image as the dots in violin plots are bigger than the textual font in the same figure.
Lines 75-83 needs to be rewritten again. The authors claim that they have isolated exosomes from all cell types within the GBM TME but they strongly focus/convey saying GBM exosomes instead of saying GBM TME derived Exosomes.
How does the goal of identifying GBM specific markers on exosomes will help to improve the isolation of circulating EVs in patients ? The authors needs to clarify this.
The authors have not isolated circulating EVs/exosomes from GBM patients (from serum/plasma/CSF/Urine) no they should be really careful about saying these terms in this particular paragraph.
Fig 5 lacks statistical evaluation. The data looks good but needs to strengthen it with significance values.
The authors needs to get familiarise with the current MISEV guidelines 2018 and 2023 and cite it in their manuscript. This will help to strengthen the paper with more robust data.
The identification of markers from the GBM patients needs to be validated in immortal cell lines which are IDH WT. This will help many researchers to compare the patients data with cell lines and use it in their study.
The authors only focussed on the multiplex assay that determines the tetraspanins proteins on the exosomes, but as suggested earlier, it would be ideal to evaluate the same with Size exclusion chromatography with 3-5 patients samples and see if they can get similar data. Otherwise identifying proteins with one techniques and saying that works as biomarker doesn’t work in general practices. Validation with another technology will definitely strengthen the results.
The authors failed to use Exosomes as control from Brain endothelial cells and Human astrocytes and compare the proteins with exosomes from GBM tissues. This is very important and needs to be included in the manuscript.
Author Response
Dear Reviewer,
We would like to express our sincere gratitude for your thorough review of our manuscript and for the insightful observations you provided. Your comments have been invaluable in guiding us to refine and strengthen the scientific rigor of our work, and we have given careful consideration to each of your suggestions.
In the attached document, we have included detailed responses to all of your observations, and the corresponding revisions have been implemented and clearly highlighted in the revised files. For ease of reading, we have formatted your comments in bold, followed directly by our responses.
We hope these modifications meet your expectations and enhance the overall quality of the manuscript, making it suitable for publication in Cancers.
We greatly appreciate your dedication to improving our work and would be pleased to address any further suggestions you may have.
Thank you once again for your time and valuable insights.

Reviewer 2 Report
Comments and Suggestions for Authors
The work presented by Franceschi et al is a relevant contribution to GB study, in which tumor explants are investigated in culture conditions in order to assess exosome release, exosome markers and the comparison of findings with publicly available datasets, in order to assess correlations with survival, cell origin and better understand the interplay of tumor cells with other TME components. I personally see the work as original and well conducted, bringing novelty with methods for exosome obtention and analyses.
Some questions/suggestions below:
1. Why only IDH wt and 1p19q deletion samples were chosen? Were cases from public databases also chosen with the same criteria? Otherwise, it may introduce some noise in comparisons. Explanations about this point might appear in the manuscript.
2. Section 2.1 describes sample collection, and section 2.2 describes how samples are selected. However, it is not clear how much time passes in this period, thus postmortem changes could certainly take place in samples. Can we discard fast postmortem changes occurring in this time frame? I understand that there is no easy solution for this, but we might at least recognize that there is no fast, snap frozen storage.
3. Figure 1B shows the IHC features of the explant after 1 week after of culture, but this is only part of the information. Did authors perform a comparison at the culture starting point vs 1 week after culturing? In my opinion, this is the comparison that allows to infer that no changes appear under culture. Still, an estimation on how much time these cultures are feasible to maintain would be interesting for readers.
4. As per the protein concentration mentioned (line 253 and supplementary figure 1), there is a lot of variability between samples, as large as 1 magnitude order. Authors might mention average and standard deviation, and discuss about the observed variability. Is there some relationship between protein concentration and maybe tissue necrosis?
5. The expression of some surface studied markers (line 291-292, and figure 2B) was even lower than the negative controls. How was this negative control chosen and why does it give more signal than other studied markers?
6. Regarding figure 3 (also lines 354-355), an explanation about variables represented in PC1 and PC2 is needed. Which variables explained the highest % of variability? I missed information about variables most represented in PC1 and PC2.
7. "This collective increase in biomarker expression in GBM suggested the potential of these markers as a collective signature of the disease" (lines 351-352). I don’t fully agree with this statement at this point. Yet it is true that 3 out of the 9 genes presented a correlation with survival, there are challenges or conceptual gaps associated with that. First, from my point of view it is unlikely that enough tumor-derived exosomes are obtained in a consistent way for evaluation in patients during disease evolution, and generating explants each time does not sound feasible. Even in case it was possible, my further question is what will doctors do with this information? Patient will probably die faster, but at this moment, this is all. A promising signature would be one that pointed to patients that would benefit from a given therapy (not first line), maybe patients that would benefit from immunotherapy, so allowing for an informed decision regarding treatment. This part should be streamlined in the sense that is indeed interesting but at this point, this type of information would not really imply a change in patient management (in addition to the point that authors recognize that the cohort is too small to represent population variability). This is partially addressed in discussion but still, maybe stressing that the novelty of the method could open the way for new knowledge that could represent a change in patient management.
8. In figure 4, why some correlation values have different font sizes? I think it is not needed and introduces crowd in the figure. If you wanted to indicate that these have more significant values, use colors and also indicate in figure legend the cut-off value.
9. In line 546, authors state that "it will be crucial to compare exosome data from GBM patients with that from healthy controls". I think this is not really the point, since it already became clear that there are differences. An interesting comparison (although more difficult to achieve) would be to compare samples from a first resection prior to treatment, with samples from patients with regrowth/relapse. What is driving relapsing? Can it be detected in advance? This would be relevant. My suggestion is to discuss feasibility/interest of this possible future investigation.
Author Response

(The authors gave the same response as above.)

Round 2
Reviewer 1 Report
Comments and Suggestions for Authors
Dear Authors,
Thank you for submitting the revised verison. The new version of manuscript have improved significantly and I would be happy to tell editors to accept it in the present form.
Thank you for your submission.